# Prognostic significance of severe coronary microvascular dysfunction post-PCI in patients with STEMI: A systematic review and meta-analysis

**Marjorie Canu[1], Charles Khouri[2,3], Stéphanie Marliere[1], Estelle Vautrin[1], Nicolas Piliero[1], Olivier Ormezzano[1,4], Bernard Bertrand[1], Hélène Bouvaist[1], Laurent Riou[4], Loic Djaileb[4], Clémence Charlon[1,4], Gerald Vanzetto[1,4,5], Matthieu Roustit[2,3], Gilles Barone-Rochette[1,4,5]***

1 Department of Cardiology, University Hospital, Grenoble-Alpes, France, 2 INSERM U1042, HP2 Laboratory, University Grenoble-Alps, Grenoble, France, 3 Clinical Pharmacology Department, INSERM CIC1406, University Hospital, Grenoble, France, 4 INSERM U1039, Bioclinic Radiopharmaceutics Laboratory, Grenoble, France, 5 French Clinical Research Infrastructure Network, Paris, France

* gbarone@chu-grenoble.fr

**Data Availability Statement:** All relevant data are within the manuscript and its Supporting Information files.

## Abstract

Coronary microvascular dysfunction (CMVD) is common and associated with poorer outcomes in patients with ST Segment Elevation Myocardial Infarction (STEMI). The index of microcirculatory resistance (IMR) and the index of hyperemic microvascular resistance (HMR) are both invasive indexes of microvascular resistance proposed for the diagnosis of severe CMVD after primary percutaneous coronary intervention (pPCI). However, these indexes are not routinely assessed in STEMI patients. Our main objective was to clarify the association between IMR or HMR and long-term major adverse cardiovascular events (MACE), through a systematic review and meta-analysis of observational studies. We searched Medline, PubMed, and Google Scholar for studies published in English until December 2020. The primary outcome was a composite of cardiovascular death, non-cardiovascular death, non-fatal myocardial infarction, non-fatal stroke, and rehospitalization for heart failure occurring after at least 6 months following CMVD assessment. We identified 6 studies, reporting outcomes in 1094 patients (mean age 59.7 ± 11.4 years; 18.2% of patients were women) followed-up from 6 months to 7 years. Severe CMVD, defined as IMR > 40 mmHg or HMR > 3mmHg/cm/sec was associated with MACE with a pooled HR of 3.42 [2.45; 4.79]. Severe CMVD is associated with an increased risk of long-term adverse cardiovascular events in patients with STEMI. Our results suggest that IMR and HMR are useful for the early identification of severe CMVD in patients with STEMI after PCI, and represent powerful prognostic assessments as well as new therapeutic targets for clinical intervention.

**Funding:** The author(s) received no specific funding for this work.

**Competing interests:** The authors have declared that no competing interests exist.

## Introduction

Primary percutaneous coronary intervention (PCI) is widely recognized as the most effective reperfusion modality for ST Segment Elevation Myocardial Infarction (STEMI). Despite the high rate of success of routine coronary reperfusion, impaired myocardial reperfusion due to coronary microvascular dysfunction (CMVD) affects half of all patients with acute myocardial infarction (MI), and represents an adverse prognostic factor [1]. CMVD is due to multiple mechanisms during myocardial infarction: distal embolization of the thrombus, endothelial dysfunction, reperfusion injury, and intramyocardial hemorrhage [2]. Consequently, better risk-stratification tools are required to determine patients with myocardial reperfusion failure who could benefit from more intensive therapeutic strategies to improve perfusion.

Standard clinical methods for assessing the efficacy of reperfusion, such as electrocardiogram and angiographic perfusion grade, are indirect and have limited diagnostic accuracy for culprit artery microvascular function [1]. Microvascular damage revealed noninvasively by contrast-enhanced cardiac magnetic resonance (CMR) imaging is an independent predictor of prognosis after MI [3]. However, CMR is usually performed several days after MI, which is too late to allow early intervention for the prevention or treatment of severe microvascular damage. In addition, CMR has limited availability in routine practice. Therefore, a direct, quantitative measure of microvascular function during primary PCI would be interesting for very early risk stratification in routine clinical practice.

Several invasive measures of microvascular function available at the time of emergency PCI [4], the index of microcirculatory resistance (IMR) or the index of hyperemic microvascular resistance (HMR) are the most used and tested in STEMI patients. These indexes use an intra-coronary guidewire to estimate microvascular resistance from simultaneous distal coronary artery measurements of pressure and flow during hyperemia, either by thermodilution for IMR, or Doppler for HMR [5]. Several studies have reported an independent association between severe CMVD at the acute phase of MI and major adverse cardiovascular events (MACE). However, the prognostic value of IMR and HMR remains unclear, mostly due to small sampling size and short-term follow-up in these studies. Therefore, the diagnosis of severe CMVD is not implemented in the routine management of STEMI yet. In order to clarify the value of invasive indexes of microvascular resistance, the main objective of this work was to conduct a systematic review of the existing literature related to the clinical implications of severe CMVD in STEMI patients. We subsequently performed a meta-analysis of available studies to better estimate the risk of cardiovascular events associated with severe CMVD in these patients.

## Methods

We conducted a systematic review and meta-analysis of cohort studies reporting associations between CMVD at baseline in patients with STEMI due to obstructive coronary artery disease (CAD) and the risk of cardiovascular events over a 6-month period, or longer. It was conducted according to the Preferred Reporting Items for Systematic Reviews and Meta-analyses (PRISMA) guidelines [6] and the latest guideline for prognostic factor meta-analysis [7]. PRISMA checklist is presented in S1 Checklist. This systematic review and meta-analysis were registered at the International Prospective Register of Systematic Reviews (PROSPERO; No.: CRD42019129380).

### Study selection

We searched Medline, PubMed, and Google Scholar by medical subject heading through December 2020, using a predefined search strategy. Exact search terms are available as [S1 File].

Two authors (MC and GBR) screened all articles and selected those meeting the pre-specified inclusion and exclusion criteria. We resolved disagreements by consensus with a third investigator (MR). All studies included in this meta-analysis met the following inclusion criteria: inclusion of patients presenting with STEMI due to obstructive CAD; CMVD assessed with HMR or IMR after PCI; occurrence of MACE, defined as a composite of two or more among the following events: all-cause death, cardiovascular death, non-fatal myocardial infarction, non-fatal stroke, and hospitalization for heart failure, assessed over a period of 6 months or longer (primary outcome). **Exclusion criteria** were: non-English language; non-human studies; reviews, editorials, abstracts; diagnostic accuracy studies without clinical outcomes; follow-up duration < 6 months; studies enrolling patients with CMVD other than post PCI STEMI (cardiomyopathies, heart-transplant recipients); studies with insufficient data to estimate hazard ratios (HR) and their 95% confidence intervals (CI). According the literature, it appeared that the IMR and HMR thresholds for the diagnosis of severe CMVD related to worse prognosis varied among teams. To standardize and stay around the most represented thresholds, severe CMVD was defined by a post-PCI IMR > 40mmHg/s (+/- 10%) or a HMR > 3 mmHg/cm/sec (+/- 10%).

The following variables were independently extracted by two investigators (MC and GBR): first author's name, publication year, source of data, number of patients, median age, percentage of women, baseline coronary morbidity, method of hyperemia induction, outcome definition, follow-up duration, mean HMR or IMR, unadjusted and adjusted HR for MACE, and their 95% CI, and the variables used for adjustments.

### Quality assessment

We appraised the quality of each included study through the Quality in Prognostic Studies (QUIPS) tool [8] and that of the overall evidence of the meta-analysis through the adapted version of the Grading of Recommandations, Assessment, Development and Evaluations (GRADE) tool for prognostic factor studies [9].

### Statistical analysis

Statistical analyses were performed with R (version 3.6.1). We synthetized the results through HR and 95% CI. For one study [10], unadjusted HRs were back calculated from the survival data [11]. A pairwise meta-analysis was then performed using DerSimonian and Laird method to assess pooled HR and 95% CI in each patient group. If in a study HR was not reported and no data was usable for calculation, we thus approximated HR **through** relative risk [11]. Given the included observational data, we used a random effect model to perform the meta-analysis. We also performed a sensitivity analysis excluding studies with a high risk of bias. Lastly, we drew a funnel plot to assess publication bias. A P value < 0.05 was considered statistically significant.

## Results

### Study characteristics

A total of 6 studies met inclusion criteria and were included in the meta-analysis (Fig 1). There was a total of 1094 subjects from these studies. Mean age was 59.7 ± 11.4 years; 18.2% of patients were women. Five studies were prospective, and one was retrospective; they were all published between 2013 and 2020. Baseline characteristics of included studies [10, 12–16] are listed in Table 1.

For all studies, IMR and HMR were performed on infarct-related artery. According the 1094 patients of meta-analysis, 551 (50.5%) infarct-related arteries were left anterior

**Fig 1. Flowchart of the study selection process using PRISMA tool.** CMVD = coronary microvascular dysfunction; MACE = Major Adverse Cardiovascular Events; CABG = Coronary artery bypass graft; CAD = coronary artery disease.

descending arteries (LAD), 132 (12%) were left circumflex (LCx), 406 (37%) were right coronary arteries (RCA), and 5 (0.5%) were left main coronary artery (LM). Jin et al [10] presented a distribution was as follows: LAD 66%, LCx 6%, and RCA 28%. For Carrick et al [12], distribution was: LAD 37%, LCx 18%, RCA 43%, and LM 2%. DeWaard et al [13] had this distribution: LAD 63%, LCx 10%, and RCA 27%. The distribution of Fearon et al [14] was: LAD 55%, LCx 9%, and RCA 36%. Fukunaga et al [15] presented the distribution as follows: LAD 52%,

**Table 1. General characteristics of included studies.**

| Author and Publication year | Source of data | Number of patients | Age (SD) | % Women | Follow up (months) | All-cause death n (%) | Rehospitalization for heart failure n (%) | Non-fatal MI n (%) | Total number of events n (%) | Optimal cut-off value | Unadjusted HR (95% CI) | Adjusted HR (95% CI) |
|---|---|---|---|---|---|---|---|---|---|---|---|---|
| Jin, 2015 | RC | 145 | 56.2 (11.8) | 11% | 85 | 11 (7.6) | 14 (9.7) | N/A | 25 (17.2) | HMR>2.82 mm Hg cm−1 s | 1,62 (1.09–2.40) | 1,74 (1.35–2.26) |
| Fukunaga, 2014 | PC | 88 | 67 (13) | 17% | 6 | 6 (6.8) | 10 (11.3) | 0 (0) | 16 (18.2) | IMR > 37 | | 0,99 (0.97–1.02) |
| Carrick, 2016 | PC | 288 | 60 (12) | 27% | 27.8 | 8 (2.8) | 22 (7.6) | N/A | 30 (10.4) | IMR > 40 | 4,36 (2.1–9.06) | 4,70 (2.10–10.53) |
| Fearon, 2013 | PC | 253 | 56,8 (10,6) | 14,60% | 33.6 | 11 (4.3) | 24 (9.5) | N/A | 35 (13.8) | IMR > 40 | 2,10 (1.10–4.10) | 2,20 (1.10–4.50) |
| DeWaard, 2017 | PC | 176 | 59,5 (10,3) | 20% | 38.4 | 8 (4.5) | 9 (5.1) | N/A | 17 (9.7) | HMR ≥3.0 mmHg cm−1 s | 1,41 (1.10–1.810) | 1,55 (1.18–2.04) |
| Maznyczka 2020 | PC | 144 | 59 (11) | 20% | 12 | 3 (2) | 19 (13.2) | 1 (0.7) | 23 (16) | IMR > 40 | NA | NA |

RC = retrospective cohort, PC = prospective cohort, SD = standard deviation, HF = heart failure, MI = myocardial infarction, HMR = hyperemic microvascular resistance index, IMR = index of microcirculatory resistance, HR = hazard ratio, NA = not available.

LCx 9%, and RCA 39%. Finally, for Maznyczka et al [16], distribution was: LAD 37%, LCx 46%, and RCA 17%.

Several others surrogate for microcirculatory disease were used in these studies as Thrombolysis In Myocardial Infarction (TIMI) flow grade [10, 12–16], corrected TIMI frame count (TFC) [12–16], myocardial perfusion grade (MPG) [12, 16], and microvascular obstruction (MO) [12, 13, 15, 16].

The overall risk of bias according to the QUIPS tool is presented in Fig 2. Three studies were considered as being at high risk of bias [9, 14, 15].

## Association between CMVD and cardiovascular events

On meta-analysis, severe CMVD was significantly associated with an increased risk of MACE, with a pooled HR of 3.42 [2.45; 4.79] (Fig 3). Sensitivity analyses were conducted after removing studies at high risk of bias. Omitting these studies did not affect the pooled HRs significantly (Fig 4).

## Risk of bias

Each study was considered to have adequate methodological quality but overall quality was low according to the GRADE tool (Fig 5). A funnel plot was used to detect possible publication bias. No asymmetry in the funnel plot was visually detected (Fig 6). Yet, the limited number of studies included in the meta-analysis was too low to interpret the results with confidence or to perform statistical tests of asymmetry.

## Discussion

To our knowledge, the present study is the first systematic review with meta-analysis assessing the association between severe CMVD and prognosis in patients with STEMI. Our results suggest that the presence of severe CMVD is a significant predictor of MACE in these patients.

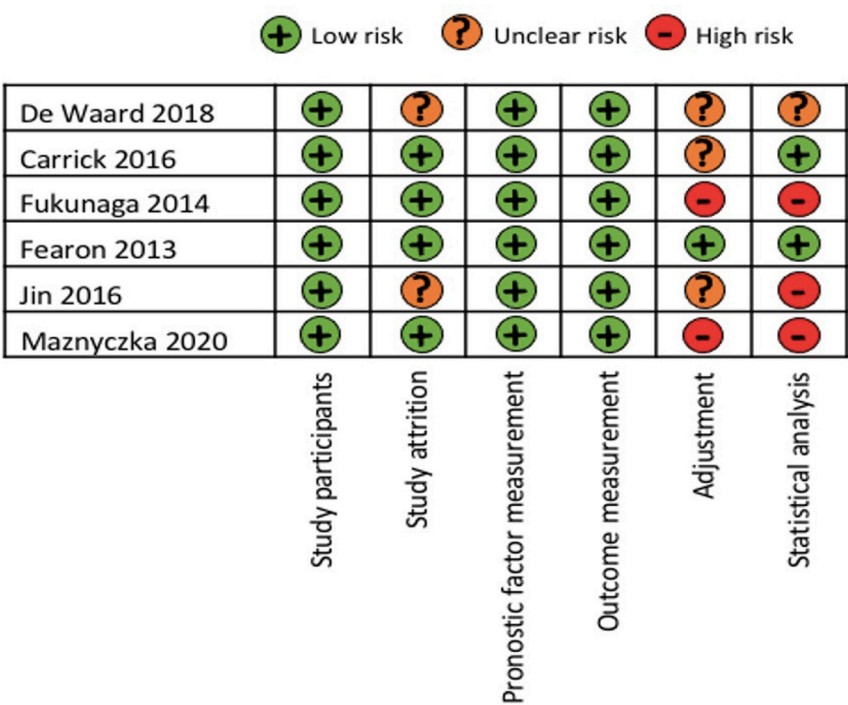

**Fig 2. Quality of included studies using QUIPS tool.** QUIPS = Quality in Prognostic Studies.

Non-invasive tests for assessing CMVD, including CMR, may be performed too late to implement potential therapies to minimize its deleterious effect. Moreover, it was reported that discordant prognostic data were obtained from IMR and MVO in nearly one-third of cases [17]. Measuring invasive indexes of microvascular function during primary PCI thus shows several advantages. However, conventional invasive methods, such as Thrombolysis In Myocardial Infarction (TIMI) flow grade, corrected TIMI frame count, and TIMI myocardial perfusion grades have important limitations: such methods depend on resting hemodynamics, are semiquantitative, and their results are not independent of the epicardial vessel. On the other hand, IMR and HMR are invasive methods which assess cardiac microvasculature quantitatively and reproducibly and in an independent way to the epicardial vessel unlike coronary flow reserve [18, 19]. Studies have shown that IMR and HMR measured during optimal hyperemia are not affected by the coronary hemodynamic status such as heart rate, preload, or afterload [20, 21]. Our results further suggest that these methods could be valuable prognostic tools to implement in routine practice. It should be noted that in the studies presented in this meta-analysis other potential surrogates for microcirculatory disease (TIMI, TFC, MPG myocardial perfusion grade, MVO) were used but appeared to provide lower prognostic performance in comparison to IMR or HMR.

IMR and HMR are derived from Ohm's law: resistance = pressure gradient/ flow. Under maximal hyperaemia, venous pressure is assumed to be zero, and therefore the pressure gradient across the microcirculation can be taken as the distal epicardial coronary artery pressure (Pd). The difference between IMR and HMR is explained by the measurement of coronary flow. IMR estimates flow with thermodilution, whereas HMR incorporates Doppler flow velocity. Each technique has inherent theoretical assumptions with its own limitations. Finally, absolute coronary blood flow (ml/ min) is not measured. Based on the law of thermodilution

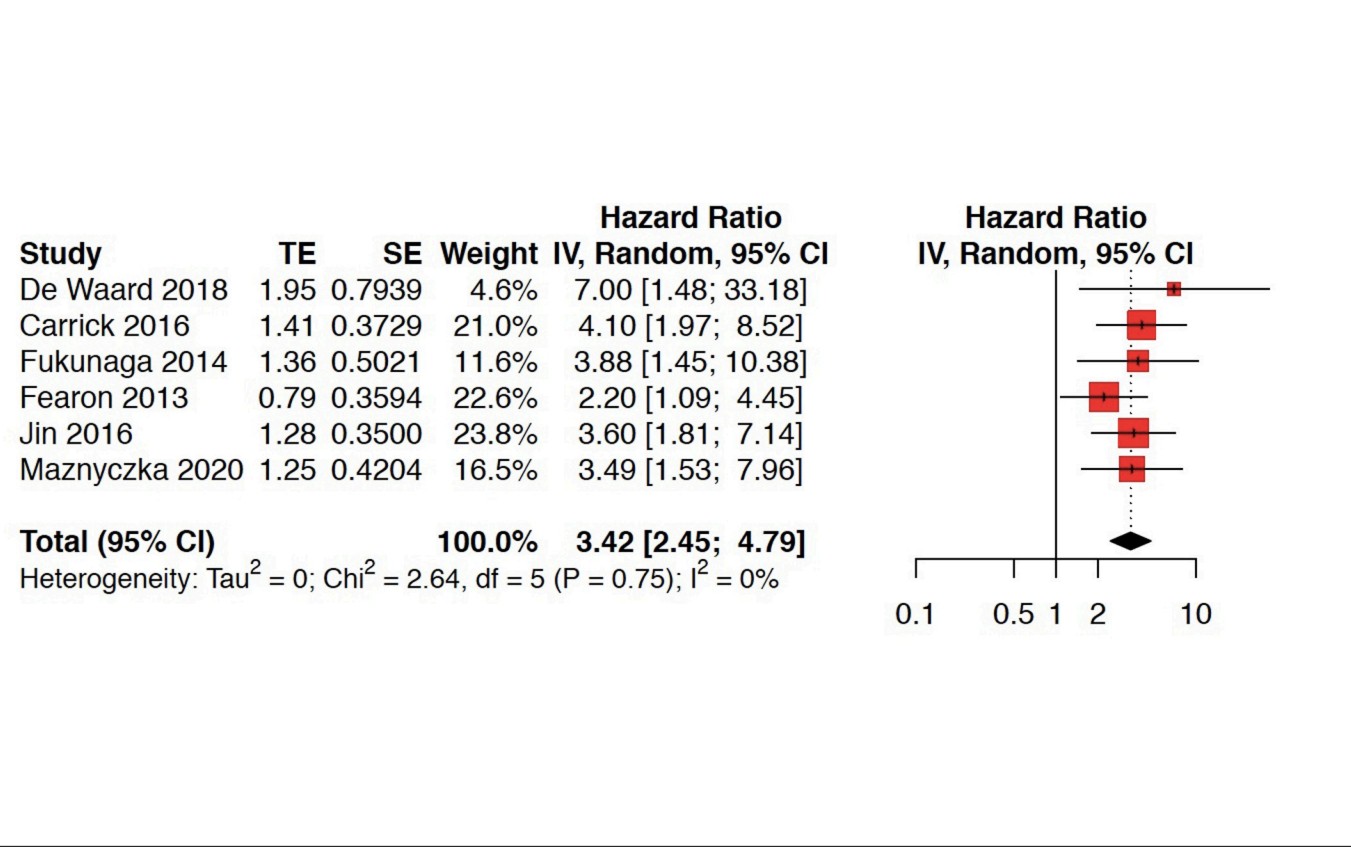

**Fig 3. Forest plot of MACE with and without severe CMVD.** CMVD was significantly associated with an increased risk of MACE, with a pooled HR of 3.42 [2.45; 4.79]. Squares and diamonds = hazard ratios, lines = 95% CI. TE = treatment effect; SE = standard errors; MACE = Major Adverse Cardiovascular Events; CMVD = coronary microvascular dysfunction; CI = confidence interval.

model, flow can be calculated from the mean time it takes a fixed vascular volume to travel from an injector to a sensor. De Bruyne et al [22] and Pijls et al [23] applied the thermodilution technique in an experimental model and found a strong correlation between the inverse of Tmn (1/Tmn) and absolute coronary flow. However, absolute coronary flow is not equal to 1/Tmn and a limitation of IMR consists in the potential variability associated with manual saline injections or thermistor position in vessel. For HMR, again based on the indicator-dilution principle, since velocity is proportional to flow, the HMR is defined as the ratio of Pd to the average peak velocity (APV) measured during hyperaemia. APV at maximal hyperemia is used as surrogate of absolute coronary blood flow. The assessment of HMR is probably more challenging than that of IMR, with higher failure rates related to unreliable doppler flow velocity tracings [24]. Therefore, in the absence of techniques allowing the direct visualization of the coronary microvasculature, the evaluation of microvascular abnormalities in clinical practice requires a surrogate with unavoidable limitations yet allowing the identification of severe CMVD using specific cut-off values.

Another issue lies in the plurality of cut-offs that have been used so far to define CMVD. Patients with IMR ranging between 32 to 40 mmHg/s presented significantly higher peak creatine kinase (CK) concentrations, no improved echocardiography-derived wall motion scores at 3 months [25, 26] and higher infarct size and MVO by CMR [27, 28]. Another study has

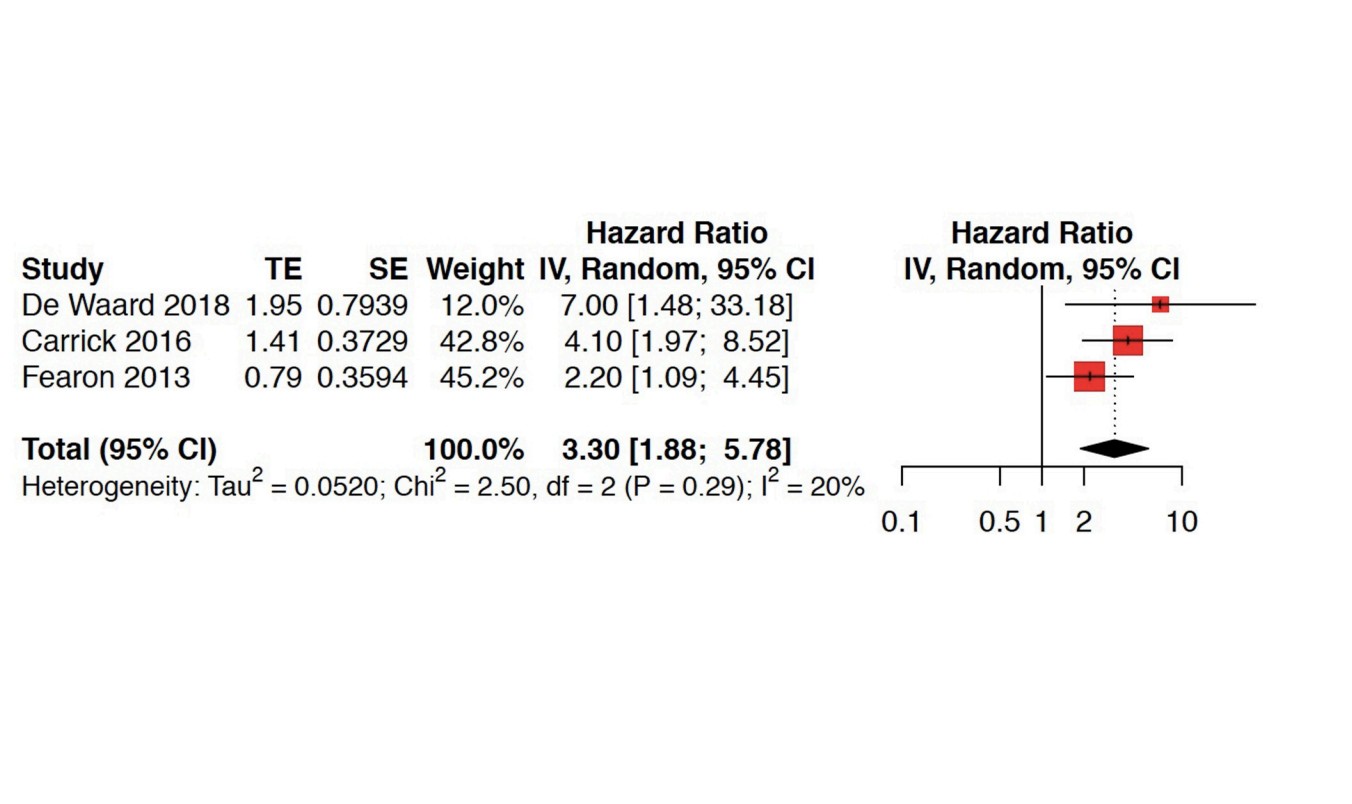

**Fig 4. Sensitivity analyses.** Sensitivity analyses consisted in repeating the forest plot after removing 3 studies at high risk of bias.

shown that an IMR > 40 (severe CMVD) is significantly associated with an increased risk of death or rehospitalization for congestive heart failure (HF) at 2.8 years [13]. Carrick et al have shown that the respective IMR cut-offs relevant for infarct pathologies and adverse health outcomes differed slightly. An IMR of 27 was most closely associated with MVO and myocardial hemorrhage, whereas a higher IMR of 40 was most closely associated with all-cause death or HF [29]. As for HMR, the cut-offs adverse health outcomes in acute coronary syndromes is not clearly defined.

## Clinical implications

IMR and HMR can allow an early risk stratification and the systematic assessment of CMVD after PCI should be generalized to provide high risk patients closer long-term monitoring to identify for example the development of heart failure and prompt treatment. Currently the obstacle of systematization are multiples. First, the assessment of IMR and HMR involves a specific diagnostic guidewire and the use of intravenous adenosine which may lengthen the procedure by $\approx$ 5 minutes. The efficiency of the IMR / HMR guidewires might turn out to be less efficient than that of guidewires classically used for PCI, which may imply the use of several guides in a thrombotic environment. In addition, adenosine can cause hemodynamic side effects (second- or third-degree heart block, hypotension, etc). However, the US Food and Drug Administration issued a safety announcement on the risk of MI and death in patients

| Study limitations | Inconsistency | Indirectness | Imprecision | Publication bias | Moderate/large effect size | Dose effect | Overall quality |
|---|---|---|---|---|---|---|---|
| ✕ | ✓ | ✕ | ✓ | ✕ | ✓ | ✕ | Low |

GRADE factors: ✓, no serious limitations; ✕, serious limitations (or not present for moderate/large effect size, dose effect); unclear

**Fig 5. GRADE framework.** Quality of evidence using GRADE framework adapted to prognostic factor studies. GRADE = Grading of Recommandations, Assessment, Development and Evaluations.

receiving adenosine for stress testing, and a prospective, multicenter study has demonstrated its safety in patients with acute or recent MI [30]. On the other hand, these minor technical constraints were able to drive a selection bias explaining that the mortality rate in studies was lower than in other reports of patients with STEMI, suggesting that the patients included in these studies may represent a lower-risk population.

If the therapeutic window for treatment of severe CMVD after STEMI occurs immediately after reperfusion [31] and that IMR and HMR can allow an early risk stratification, unfortunately even these measures are good for prognostication, currently there is no different treatment that can modify the outcomes. Additional studies are needed to establish specific new therapeutic strategies using HMR or IMR for the identification of patients at increased risk after STEMI. Indeed, IMR or HMR are theragnostic biomarkers, i.e. **they are a metric** that predicts therapeutic response. They could be tools for the development of a stratified medicine and go towards a precision medicine. Because CMVD does not occur in all patients with STEMI, the design of such studies could be improved by the initial selection of patients at higher risk, in whom adjunctive treatment could actually be beneficial. Several studies are in progress with the use of IMR as a theragnostic biomarkers for a stratified medicine. Drugs (intracoronary fibrinolytics or vasodilator), devices (pressure-controlled intermittent coronary sinus occlusion) or PCI strategies (deferred stenting) are currently being evaluated [5]. Interestingly, the results of these studies will lead to a new debate on therapeutic cut-offs, as they will be lower than prognostic cut-offs. In the RESTORE-AMI study (NCT03998319), only patients with an IMR >32 will be eligible for randomization between intracoronary tenecteplase and placebo. In the OPTIMAL study (NCT02894138), IMR >30 was chosen to select and randomize STEMI patients between intracoronary alteplase and placebo.

## Limitations

Several limitations should be pointed out. First, there was a limited number of eligible studies. Moreover, all studies included in this meta-analysis were observational, with relatively small sample sizes. Finally, our meta-analysis used pooled data rather than individual patient data, which restricted the detailed analysis of potential confounding factors such as patient characteristics, medical treatments, PCI procedural, and follow-up period. Despite these limitations, statistical heterogeneity was minor, and the main result was unchanged following the exclusion of studies at high risk of bias. Also, we decided to include both IMR and HMR in this meta-analysis while these two invasive indexes of coronary microvascular resistance cannot be considered equivalent. The correlation between these two indexes is indeed far from being strong (rho = 0.41, p < 0.0001) [32]. This could represent a limitation for our results interpretation. To overcome this issue, we ran a subgroup analysis by splitting the studies using IMR

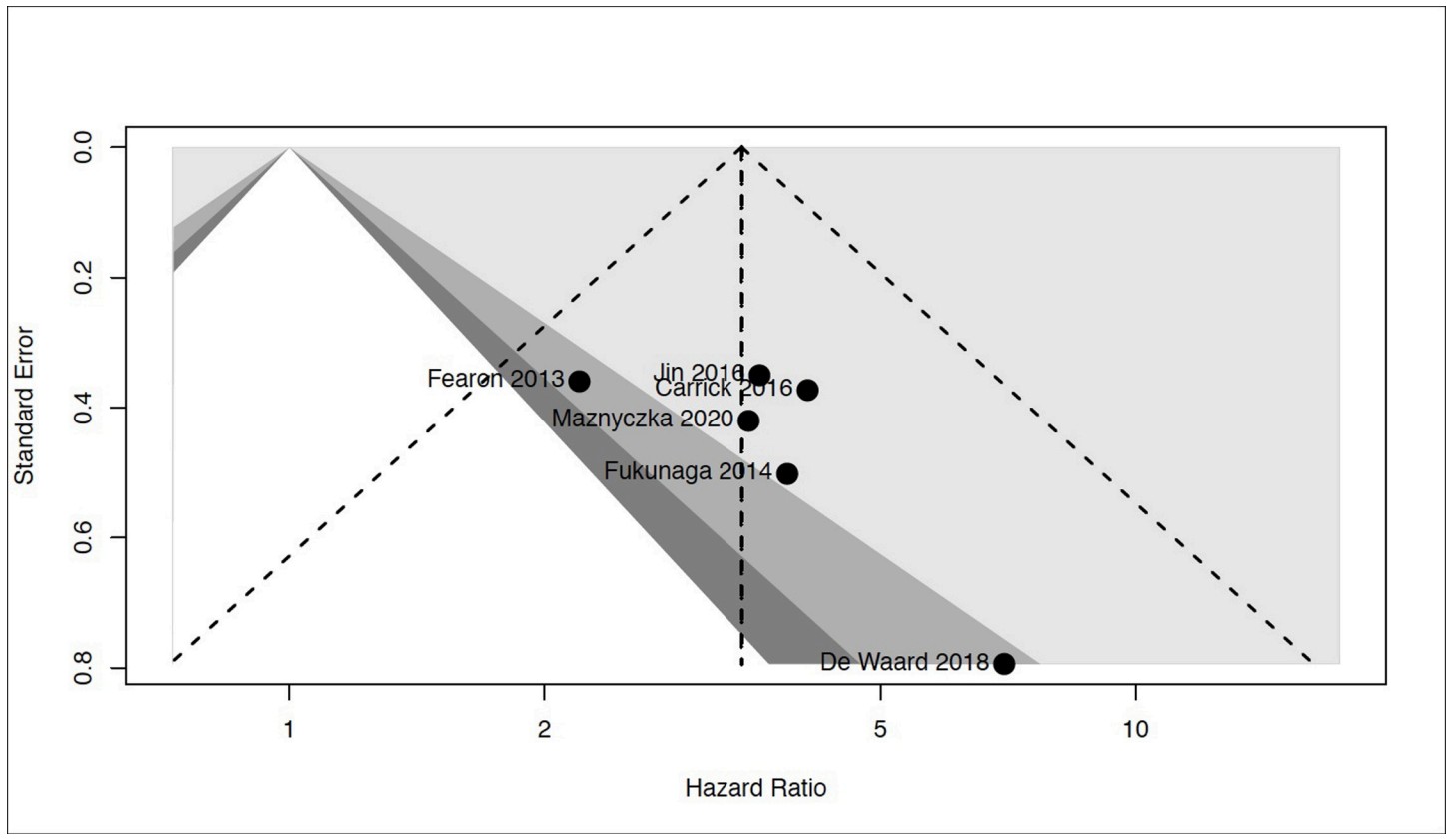

**Fig 6. Funnel plot.** The funnel plot showed no asymmetry, suggesting no publication biais.

and HMR, and there was no difference between the two methods to assess CMV and the outcomes of interest (p = 0.56) [S1 Fig].

## Conclusion

Our results show that severe CMVD assessed during PCI is associated with adverse cardiovascular events in patients with STEMI. Considering the risk of events in these patients, the early identification of severe CMVD in patients with acute STEMI after PCI by IMR and HMR could represent a powerful prognostic tool. Further research is needed to investigate whether severe CMVD could be used as therapeutic target for preventive or therapeutic interventions in patients stratified by IMR or HMR.

## Supporting information

**S1 Checklist. PRISMA checklist.**
(DOC)

**S1 File. Exact search terms used for systematic reviewing in Medline, Pubmed and Google scholar.**
(DOCX)

**S1 Fig. Forest plot of MACE with and without severe CMVD splitting IMR and HMR studies.** There was no difference between the two methods to assess CMV and the outcomes of

interest (p = 0.56).
(TIFF)

## Author Contributions

**Conceptualization:** Marjorie Canu, Charles Khouri, Matthieu Roustit, Gilles Barone-Rochette.

**Formal analysis:** Marjorie Canu, Charles Khouri, Matthieu Roustit, Gilles Barone-Rochette.

**Funding acquisition:** Marjorie Canu, Charles Khouri, Matthieu Roustit, Gilles Barone-Rochette.

**Methodology:** Marjorie Canu, Charles Khouri, Matthieu Roustit, Gilles Barone-Rochette.

**Validation:** Marjorie Canu, Charles Khouri, Matthieu Roustit, Gilles Barone-Rochette.

**Writing – original draft:** Marjorie Canu, Charles Khouri, Matthieu Roustit, Gilles Barone-Rochette.

**Writing – review & editing:** Marjorie Canu, Charles Khouri, Stéphanie Marliere, Estelle Vautrin, Nicolas Piliero, Olivier Ormezzano, Bernard Bertrand, Hélène Bouvaist, Laurent Riou, Loic Djaileb, Clémence Charlon, Gerald Vanzetto, Matthieu Roustit, Gilles Barone-Rochette.

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
