## [Decision Letter · Decision Letter 0]

15 Feb 2022

PONE-D-21-31414

Prognostic significance of severe coronary microvascular dysfunction post-PCI in patients with STEMI: a systematic review and meta-analysis

PLOS ONE

Dear Dr. Barone-Rochette,

Thank you for submitting your manuscript to PLOS ONE. After careful consideration, we feel that it has merit but does not fully meet PLOS ONE’s publication criteria as it currently stands. Therefore, we invite you to submit a revised version of the manuscript that addresses the points raised during the review process.

Please be sure to address each of the comments and suggestions made by all three reviewers in an individual fashion in the response letter. The authors are to be commended on their work.

We look forward to receiving your revised manuscript.

Kind regards,

R. Jay Widmer

Academic Editor

PLOS ONE

https://journals.plos.org/plosone/s/fileid=ba62/PLOSOne_formatting_sample_title_authors_affiliations.pdf".

2. Please, in the abstract of your manuscript, change the search dates to match those reported in the Methods section.

4. We noticed you have some minor occurrence of overlapping text with the following previous publication(s), which needs to be addressed:

- https://linkinghub.elsevier.com/retrieve/pii/S1443950619313241

-https://www.ahajournals.org/doi/10.1161/CIRCINTERVENTIONS.117.005361

- https://www.heartlungcirc.org/article/S1443-9506(19)31324-1/fulltext

- http://eprints.gla.ac.uk/118950/7/118950.pdf

In your revision ensure you cite all your sources (including your own works), and quote or rephrase any duplicated text outside the methods section. Further consideration is dependent on these concerns being addressed.

Additional Editor Comments:

The reviewers were quite favorable on this paper after some lengthy deliberations. Please address the minor comments below in an individual fashion prior to acceptance.

Reviewers' comments:

Reviewer's Responses to Questions

**Comments to the Author**

1. Is the manuscript technically sound, and do the data support the conclusions?

Reviewer #1: Yes

Reviewer #2: Yes

Reviewer #3: Yes

2. Has the statistical analysis been performed appropriately and rigorously? 

Reviewer #1: Yes

Reviewer #2: Yes

Reviewer #3: Yes

3. Have the authors made all data underlying the findings in their manuscript fully available?

Reviewer #1: Yes

Reviewer #2: Yes

Reviewer #3: Yes

4. Is the manuscript presented in an intelligible fashion and written in standard English?

Reviewer #1: Yes

Reviewer #2: Yes

Reviewer #3: Yes

5. Review Comments to the Author

Reviewer #1: Well-written article. I have some minor typographical/grammatical suggestions:

Non-inclusion criteria = exclusion criteria?

Trough = through?

Given we included = given the included?

they is a metric = they are?

Please check for dots and comma’s in tables

Reviewer #2: The authors described in this review and metaanalysis the prognostic impact of microcirculation damage measured invasively during STEMI. They conclude that both the IMR and the HMR, (2 quantifiable markers of damage to the microcirculation) considered qualitatively have a prognostic interest.

I have a few minor points to discuss or clarify in the introduction, method, discussion and limitations/

In the abstract please correct 2019 as a study published in 2020 was included in the metaanalyses.

Authors should justify the threshold value choosen for IMR and HMR to define severe CMVD.

Could the authors provide some data regarding the location of STEMI patients, the culprit artery? Were IMR and HMR measured on the infarted area?

Authors should discuss the use of other potential surrogates of microcirculation disease (TIMI flow, MVO for example) and acknowledge the lack of such angiographic or imaging data in the considered studies.

In the discussion the following sentenece should be referenced or removed "The

assessment of HMR is probably more challenging than that of IMR, with higher failure

rates related to unreliable doppler flow velocity tracings"

Reviewer #3: In their meta-analysis, Canu et al. aimed to conduct a systematic review of the existing literature related to the clinical implications of severe CMVD in STEMI patients. This study is well design, original and very interesting.

The main limitation is the limited number of eligible studies, but the authors emphasized this in their discussion. However, the study provides interesting insights that deserve to be published.

6. PLOS authors have the option to publish the peer review history of their article (what does this mean?). If published, this will include your full peer review and any attached files.

Reviewer #1: No

Reviewer #2: No

Reviewer #3: No

---

## [Author Response · Author response to Decision Letter 0]

21 Mar 2022

Response to Reviewers

Revised Manuscript with Track Changes

Manuscript

Thank you very much for the work of the reviewers and editors, which will increase the interest of the manuscript. You will find below a point-by-point answer to each of the reviewers’ and editor’s comments and the indications of changes made to the manuscript.

Reviewers' comments:

Reviewer #1: Well-written article. I have some minor typographical/grammatical suggestions:

Non-inclusion criteria = exclusion criteria?

Trough = through?

Given we included = given the included?

they is a metric = they are?

Please check for dots and comma’s in tables

Sorry for these mistakes, we have corrected them.

Reviewer #2: The authors described in this review and metaanalysis the prognostic impact of microcirculation damage measured invasively during STEMI. They conclude that both the IMR and the HMR, (2 quantifiable markers of damage to the microcirculation) considered qualitatively have a prognostic interest.

I have a few minor points to discuss or clarify in the introduction, method, discussion and limitations/

- In the abstract, please correct 2019 as a study published in 2020 was included in the metaanalyses.

We have corrected date in abstract. 

- Authors should justify the threshold value choosen for IMR and HMR to define severe CMVD.

We completed the manuscript to answer of this question: “according the literature, it appeared that the IMR and HMR thresholds for the diagnosis of severe CMVD related to worse prognosis varied among teams. To standardize and stay around the most represented thresholds, severe CMVD was defined by a post-PCI IMR > 40mmHg/s (+/- 10%) or a HMR > 3 mmHg/cm/sec (+/- 10%)”.

- Could the authors provide some data regarding the location of STEMI patients, the culprit artery? Were IMR and HMR measured on the infarted area?

For all studies, IMR and HMR were performed on infarct-related artery. According the 1094 patients of meta-analysis, 551 (50.5%) infarct-related arteries were left anterior descending arteries (LAD), 132 (12%) were left circumflex (LCx), 406 (37%) were right coronary arteries (RCA), and 5 (0.5%) were left main coronary artery (LM). Jin et al (10) presented a distribution was as follows: LAD 66%, LCx 6%, and RCA 28%. For Carrick et al (13), distribution was: LAD 37%, LCx 18%, RCA 43%, and LM 2%. DeWaard et al (14) had this distribution: LAD 63%, LCx 10%, and RCA 27%. The distribution of Fearon et al (15) was: LAD 55%, LCx 9%, and RCA 36%. Fukunaga et al (16) presented the distribution as follows: LAD 52%, ,LCx 9%, and RCA 39%. Finally, for Maznyczka et al (17), distribution was: LAD 37%, LCx 46%, and RCA 17%.

- Authors should discuss the use of other potential surrogates of microcirculation disease (TIMI flow, MVO for example) and acknowledge the lack of such angiographic or imaging data in the considered studies.

Several surrogates of microcirculation disease were used in the considered studies. 

We have added these elements in results session and discussion session: 

“Several others surrogate for microcirculatory disease were used in these studies as Thrombolysis In Myocardial Infarction (TIMI) flow grade [10, 13-17], corrected TIMI frame count (TFC) [13-17], myocardial perfusion grade (MPG) [13,17], and microvascular obstruction (MO) [13,14,16,17]”. 

“It should be noted that in the studies presented in this meta-analysis other potential surrogates for microcirculatory disease (TIMI, TFC, MPG myocardial perfusion grade, MVO) were used but appeared to provide lower prognostic performance in comparison to IMR or HMR”.

- In the discussion the following sentence should be referenced or removed "The assessment of HMR is probably more challenging than that of IMR, with higher failure rates related to unreliable doppler flow velocity tracings"

We have referenced the sentence: Barbato E, Aarnoudse W, Aengevaeren WR, et al. Validation of coronary flow reserve measurements by thermodilution in clinical practice. Eur Heart J. 2004;25:219-23

Reviewer #3: In their meta-analysis, Canu et al. aimed to conduct a systematic review of the existing literature related to the clinical implications of severe CMVD in STEMI patients. This study is well design, original and very interesting.

The main limitation is the limited number of eligible studies, but the authors emphasized this in their discussion. However, the study provides interesting insights that deserve to be published.

Thank you for these comments.

---

## [Decision Letter · Decision Letter 1]

28 Apr 2022

Prognostic significance of severe coronary microvascular dysfunction post-PCI in patients with STEMI: a systematic review and meta-analysis

PONE-D-21-31414R1

Dear Dr. Barone-Rochette,

We’re pleased to inform you that your manuscript has been judged scientifically suitable for publication and will be formally accepted for publication once it meets all outstanding technical requirements.

Kind regards,

R. Jay Widmer

Academic Editor

PLOS ONE

Additional Editor Comments (optional):

Reviewers' comments:

Reviewer's Responses to Questions

**Comments to the Author**

1. If the authors have adequately addressed your comments raised in a previous round of review and you feel that this manuscript is now acceptable for publication, you may indicate that here to bypass the “Comments to the Author” section, enter your conflict of interest statement in the “Confidential to Editor” section, and submit your "Accept" recommendation.

Reviewer #1: All comments have been addressed

Reviewer #3: All comments have been addressed

2. Is the manuscript technically sound, and do the data support the conclusions?

Reviewer #1: (No Response)

Reviewer #3: Yes

3. Has the statistical analysis been performed appropriately and rigorously? 

Reviewer #1: (No Response)

Reviewer #3: I Don't Know

4. Have the authors made all data underlying the findings in their manuscript fully available?

Reviewer #1: (No Response)

Reviewer #3: Yes

5. Is the manuscript presented in an intelligible fashion and written in standard English?

Reviewer #1: (No Response)

Reviewer #3: Yes

6. Review Comments to the Author

Reviewer #1: (No Response)

Reviewer #3: The authors have adequately addressed all the comments, thank you. This manuscript is now acceptable for publication

7. PLOS authors have the option to publish the peer review history of their article (what does this mean?). If published, this will include your full peer review and any attached files.

Reviewer #1: No

Reviewer #3: **Yes: **Thomas Bochaton

---

## [Editor Report · Acceptance letter]

3 May 2022

PONE-D-21-31414R1 

Prognostic significance of severe coronary microvascular dysfunction post-PCI in patients with STEMI: a systematic review and meta-analysis 

Dear Dr. Barone-Rochette:

I'm pleased to inform you that your manuscript has been deemed suitable for publication in PLOS ONE. Congratulations! Your manuscript is now with our production department. 

Kind regards, 

on behalf of

Dr. R. Jay Widmer 

Academic Editor

PLOS ONE